# Primary Cilium Identifies a Quiescent Cell Population in the Human Intestinal Crypt

**DOI:** 10.3390/cells12071059

**Published:** 2023-03-31

**Authors:** Blanche Sénicourt, Gabriel Cloutier, Nuria Basora, Sepideh Fallah, Andréanne Laniel, Christine Lavoie, Jean-François Beaulieu

**Affiliations:** 1Laboratory of Intestinal Physiopathology, Faculty of Medicine and Health Sciences, Université de Shebrooke, Sherbrooke, QC J1H5N4, Canada; 2Department of Pharmacology and Physiology, Faculty of Medicine and Health Sciences, Université de Sherbrooke, Sherbrooke, QC J1H5N4, Canada; 3Centre de Recherche du Centre Hospitalier Universitaire de Sherbrooke, Sherbrooke, QC J1H5N4, Canada

**Keywords:** primary cilium, Hedgehog pathway, intestinal epithelial cells, stem cells, HIEC-6 cell line, tubulin, GLI, patched, BMI1

## Abstract

Primary cilia are sensory antennae located at the cell surface which mediate a variety of extracellular signals involved in development, tissue homeostasis, stem cells and cancer. Primary cilia are found in an extensive array of vertebrae cells but can only be generated when cells become quiescent. The small intestinal epithelium is a rapidly self-renewing tissue organized into a functional unit called the crypt–villus axis, containing progenitor and differentiated cells, respectively. Terminally differentiated villus cells are notoriously devoid of primary cilia. We sought to determine if intestinal crypts contain a quiescent cell population that could be identified by the presence of primary cilia. Here we show that primary cilia are detected in a subset of cells located deep in the crypts slightly above a Paneth cell population. Using a normal epithelial proliferative crypt cell model, we show that primary cilia assembly and activity correlate with a quiescent state. These results provide further evidence for the existence of a quiescent cell population in the human small intestine and suggest the potential for new modes of regulation in stem cell dynamics.

## 1. Introduction

The primary cilium is a microtubule-based structure found at the surface of almost all mammalian cells. Over the past two decades, intense research has catapulted this tiny, previously ignored organelle into the spotlight of such fields as development, stem cells and cancer. Primary cilia are sensory organelles that monitor a growing list of extracellular signals under various conditions [1,2,3,4]. The first evidence of this was identified in renal epithelial cells, which use primary cilia as a mechanosensor to monitor fluid flow, i.e., shear stress [5]. Sensory organs, such as the retina and olfactory neurons use cilia to mediate chemosensation by targeting G-coupled photoreceptors [6,7] and odor receptors [8,9], respectively. These discoveries were quickly eclipsed by new studies identifying roles for primary cilia in critically mediating the regulation of the cell cycle and several developmental signalling pathways, most notably the Hedgehog pathway (HH) [10,11,12], but also including PDGF [13], Wnt [14,15], Notch [16] and Hippo [17,18] as well as being a regulator of cell size via mTOR [19] and involved the integration of cellular response to mitochondrial stress [20].

The list of cells displaying primary cilia was so extensive that the website dedicated to it was shut down, as it was easier to catalogue tissues in which cells did not have primary cilia, such as blood-borne cells that grow in suspension [21]. The mucosal epithelium of the gut is an even more puzzling exception, since most other epithelial cell types display primary cilia [22]. The functional unit of the small intestine, the crypt–villus axis, is composed of two functionally and physically distinct regions. The villus contains functionally differentiated cells, mostly absorptive cells that display brush borders composed of actin-based microvilli, and goblet cells that secrete mucins which are constantly renewed by the transit-amplifying cells of the crypts. These latter cells are highly proliferative and undifferentiated and are replenished by a robust stem cell population residing in the crypt [23]. While the primary cilium has been generally reported in quiescent cells [24], since the basal body from which the cilium grows needs to be converted into the centrosome required for the mitotic spindle in proliferative cells [25], it seems in fact that only cells directly under the mitotic process cannot grow a primary cilium [1,4,26]. Nevertheless, in the intestinal epithelium, the transit-amplifying crypt cells and the primordial stem cells from which they are derived appear not to be able to assemble primary cilia. Rare reports from detailed electron microscopy (EM) observations of mouse intestine have clearly established that the terminally differentiated intestinal epithelial cells of the villus, which are quiescent, do not generate primary cilia [27]. The lack of primary cilia in the normal intestinal epithelium was further confirmed by various reports [26,28,29,30] albeit their ubiquitous presence in the stroma and muscle cells of the intestinal wall [26,28,29,31]. 

Interestingly, the observation of functional primary cilia was reported in the small bowel and colon adenocarcinoma as well as some of their derived cancer cell lines [26,28,32,33], indicating that intestinal epithelial cells can display primary cilia at least under certain conditions. The presence of a primary cilium in colorectal cancer cells was correlated with the increased expression of the HH effector GLI1 and the recruitment of the SMO receptor [26], suggesting the potential for an autocrine pathway. As mentioned above, primordial stem cells, also identified as leucine-rich repeat-containing G-protein-coupled receptor 5-positive (LGR5+) cells, from which most intestinal adenocarcinomas are thought to be derived [34], do not possess a primary cilium as they are cycling. However, the intestinal crypt contains another stem cell type called reserve stem cells, tentatively identified as B-cell-specific Moloney murine leukemia virus interaction site 1-positive (BMI1+) cells, which are mostly quiescent and resistant to stress but have the capacity to regenerate the LGR5+ stem cell population after tissue injury, such as radiation [35,36,37]. In vitro, BMI1+ crypts as well as BMI1+ spheroids can be induced to upregulate LGR5 expression [38,39]. As recently reviewed, there is a certain complexity to the functional plasticity of these reserve stem cells for intestinal regeneration vs. that of primordial stem cells, but evidence supports the existence of stem cells with nonequivalent proliferation profiles that co-exist in the intestinal crypt [40,41]. In support of this, it is noteworthy that BMI1+ cells appear independent of the Wnt pathway for their proliferation in contrast to LGR5+ cells [39,42], while BMI1+ and LGR5+ cells have been shown to represent distinct populations of cancer stem cells in intestinal neoplasms [43]. In this context, we hypothesized that intestinal BMI1+ quiescent stem cells may bear functional primary cilia.

## 2. Materials and Methods

### 2.1. Human Intestinal Tissue Samples

Normal adult ileal samples were obtained from Québec Transplant (Montréal, QC, Canada) in accordance with protocols approved by the Institutional Human Subject Review Board of the Centre Hospitalier Universitaire de Sherbrooke for the use of human material. The average age of the donors was 49.5 ± 21.0 (21–82). A total of 46% of the donors were female and 54% were male. The preparation and embedding of tissues for cryosectioning were as described previously [44,45].

### 2.2. Cell Culture

The HIEC-6 cell line is a human intestinal crypt cell model. The cells are available from the American Type Culture Collection (CRL-3266, ATCC, Manassas, VA, USA). HIEC-6 cells are normal non-transformed/non-immortalized cells that were generated and grown as described previously [46,47]. HIEC-6 cells have been characterized as exhibiting morphological and functional characteristics of normal human proliferative crypt cells and are considered to be undifferentiated crypt-like progenitor cells [38,48,49,50,51]. 

In some experiments, cell synchronization was performed as follows: 5 × 10^4^ HIEC-6 cells were plated on serum-coated glass coverslips and allowed to adhere for 24 h. Cells were then serum-starved for 72 h to impede cell growth. Complete media were then added, and cell cultures were monitored with anti-acetylated tubulin (as described below in 2.3) at different time points over a period of 48 h. The number of cells in M-phase (prophase to telophase) or the number of ciliated cells were counted and expressed as a percentage of total cells. Representative results from one of three independent experiments are shown. 

The HH pathway was also investigated using the SMO agonist Purmorphamine at 2 µM (#483367-10-8 Cayman Chemical, Ann Arbor, MI, USA) and the GLI1 inhibitor GANT-61 at 5µM (#3191, Tocris via Cedarlane Corp., Burlington, ON, Canada). 

### 2.3. Indirect Immunofluorescence Staining and Confocal Imaging

Tissue cryosections embedded in OCT and cells were fixed in methanol at −20 °C or in paraformaldehyde 4% and processed as previously described [49,51,52,53]. The primary antibodies were anti-acetylated α-tubulin (6-11B-1, 1/2000, Sigma Aldrich, Oakville, ON, Canada), anti-polyglutamylated tubulin (GT335, 1/2000, AdipoGen Life Sciences, San Diego, CA, USA), anti-ARL13B (17711-1-AP, 1/1000, Proteintech, Rosemont, IL, USA and 90413h, 1/150, BiCell Scientific, Maryland Heights, MO, USA), anti-alpha6 integrin subunit (G0H3, 1/5000, BD Pharmingen, Mississauga, ON, Canada), anti-GLI1 (ab49314, 1/500, Abcam Inc, Cambridge, MA, USA), anti-GLI3C (1/500, kind gift from F J de Sauvage, Genentech, San Francisco, CA, USA [54]), anti-BMI1 (sc390443 clone F9, 1/20, Santa Cruz Biotechnology, Dallas, TX, USA and D20B7, 1/600, Cell Signaling Technology, Danvers, MA, USA), anti-E-cadherin (polyclonal, 1/50, Santa Cruz Biotechnology, Dallas, TX, USA), anti-group II phospholipase A2 (1/1000; kind gift from TJ Nevalainen, [55]), anti-pericentrin (ab4448, 1/200, Abcam) and anti MKLP1 (ab174304, 1/200, Abcam). Secondary antibodies used were anti-mouse, rat and rabbit Alexa 594 or 488 (Invitrogen Molecular Probes, ThermoFisher, Mississauga, ON, Canada). Nuclei were stained with DAPI (Molecular Probe). Images were acquired using a DFC300FX color camera or an RTE/CCD Y/Hz-1300 camera controlled using MetaMorph software (Universal Imaging Corporation, Downingtown, PA, USA). In some cases, selected acquired stacked images were submitted to deconvolution (Volocity, Quorum Technology, Puslinch, ON, Canada) and used to generate a 3D reconstructed image (Imaris, Oxford Instruments, Concord MA, USA). Stained tissues and cells were also viewed with an inverted confocal laser scanning microscope (FV1000; Olympus, Tokyo, Japan) equipped with a PlanApo 60×/1.42 oil immersion objective and Olympus FluoView version 1.6a to acquire and analyze the images, or with a Leica TCS SP8 STED DMI8 scanning confocal microscope (Leica Microsystems, Toronto, ON, Canada) equipped with a 63×/1.4 oil-immersion objective and a tunable white light laser (470 to 670 nm). LAS AF Lite software (Leica) was used for image acquisition and analysis. The images were further processed using Adobe Photoshop (Adobe Systems, San Jose, CA, USA).

### 2.4. RNA Extraction, Reverse Transcriptase and Quantitative RT-PCR

RiboZol (AMRESCO, Solon, OH, USA) was used for cell lysing. RNA extraction, reverse transcription and quantitative polymerase chain reaction (qPCR) assays were performed as described previously for both cells and tissues [56]. SYBR Green Power PCR Master Mix (Bio Basic, Markham, ON, Canada) was used for qRT-PCR. The primers used for qPCR included the following: *GLI1*: forward 5′-ACATCAACTCCGGCCAATAG-3′ and reverse 5′-GAGGATGCTCCATTCTCTGG-3′; *SMO*: forward 5′-CCCAGCATGTCACCAAGATG-3′ and reverse 5′-GCACACCTCCTTCTTCCTCT-3′; *PTCH1*: forward 5′-ACATCAACTCCGGCCAATAG-3′ and reverse 5′-GCCAGAATGCCCTTCAGTAG-3′; and *BMI1*: forward 5′-TGTTCGTTACCTGGAGACC-3′ and reverse 5′-CAGCATCAGCAGAAGGATG-3′. 

Gene expression was calculated according to the Pfaffl equation [57] using RPLPO as a validated normalizer [56] relative to control groups as specified in the text.

### 2.5. 5-Bromo-2′-Deoxyuridine (BrdU) Incorporation Assay

BrdU incorporation and staining were performed according to the manufacturer’s (ThermoFisher Scientific, Ottawa, ON, Canada) instructions. Briefly, cells were plated on serum-coated glass coverslips and allowed to adhere for 24 h. At the time of experimentation, cells were incubated with normal medium containing the BrdU solution for 4 h. Cells were then washed and processed for anti-BrdU and DAPI staining as described previously [50].

### 2.6. Statistical Analysis

Data preparation and statistical analyses, which included two-tailed Student’s *t*-test and ANOVA, were performed with Graph Pad Prism 8.3 (Graph Pad Software, San Diego, CA, USA). A *p* value < 0.05 was considered significant in all analyses. All experiments were repeated at least three times, independently.

## 3. Results

### 3.1. Search for Primary Cilia in the Reserve Stem Cells of the Human Intestinal Crypt

Previous studies from various laboratories, including our own, have failed to consistently identify primary cilia in the normal intestinal epithelium although their transformed counterparts exhibit primary cilia both in situ and in cellulo [26,28,29,32,33]. However, because quiescent reserve stem cells are likely to be relatively rare in the intestinal crypt in light of the complexity of the various cell populations that can contribute to epithelial regeneration [40,41], we hypothesized that the detection of epithelial crypt cells bearing a primary cilium should be a relatively infrequent event. Indeed, several crypts had to be examined to identify primary cilia in specimens of normal adult small intestinal sections stained for acetylated-α-tubulin, a marker for primary cilia which are readily identifiable by their overall intensity, shape and location using indirect immunofluorescence [58,59,60]. As confirmed by co-staining for E-cadherin, positive staining for acetylated-α-tubulin was found in the apical domain of some epithelial cells of the crypt, where a primary cilium can be distinguished from other neighboring structures by its size, intensity and localization in the lower third of the crypt (Figure 1A, arrow vs. arrowheads). The frequency of primary cilia was evaluated to be approximately one per crypt based on the examination of hundreds of crypts (i.e., one cilium detected per 25–30 crypts on 3 µm thick cryosections, with the average crypt being 50 µm). To confirm that the cells bearing the primary cilia were not Paneth cells, specimens of normal small intestine were double-labelled for the detection of acetylated-α-tubulin and group II phospholipase A2, a Paneth cell marker [55]. Indirect co-immunofluorescence confirmed that primary cilia were detected in cells near the crypt bottom but always above the Paneth cell population (Figure 1B,B’). As suggested by [1] and also considering the greater difficulty in identifying primary cilia components in tissues compared with cells in culture [61], we used the small GTPase ARL13b as a second marker for the primary cilium [62,63] to validate our observations.

As shown in Figure 2, confocal images for dual acetylated-α-tubulin and ARL13b detection confirmed the presence of a primary cilium at the base of the small intestinal crypt but also showed that the anti-ARL13B 17711-1-AP antibody identified extra dots in intestinal crypt cells more weakly or not at all stained for acetylated-α-tubulin.

The presence of the primary cilium in intestinal crypt cells was also confirmed by dual polyglutamylated tubulin [65,66] and ARL13b detection (Figure 3).

As mentioned above, BMI1 has been recognized as one potential marker for identifying reserve stem cells in the intestinal crypt. Using tissue sections from various donors, BMI1-positive clusters of two to five cells were observed in the lower parts of the small intestinal crypts, while some of these cells were consistently also stained for ARL13b detection in co-staining (Figure 4), suggesting that the crypt cells bearing a primary cilium also expressed higher levels of BMI1. As indicated in Figure 2 and Figure 3, it is noteworthy that some of the structures stained with the 17711-1-AP antibody were not always identified as a primary cilium, consistent with the finding that ARL13B has non-cilia functions, such as reported in epithelial cells [67]. Based on our previous observations, one should expect only one primary cilium per crypt on average. The nature of the other 17711-1-AP-positive structures (arrowheads in Figure 4A,B,E) could be related to midbody remnants, a type of structure reported in polarized cells and proposed to be involved in the formation of primary cilia [64]. Interestingly, the model proposed would imply that only cells with a midbody remnant can assemble primary cilia [64], supporting the idea that at least some intestinal BMI1-positive cells can become ciliated. On the other hand, some of the other positive structures were observed all along the luminal aspect of the epithelium (Figure 4F), suggesting that they may be related to the centrosome according to a pattern similar to that reported in mouse intestine for centrin2 expression [30].

To further investigate the expression of the primary cilium in the human intestinal crypts in relation to midbody remnants and centrosomes, another antibody raised against the C-terminal sequence of ARL13B (hereafter called the 90413h antibody) was tested in double staining for the detection of pericentrin, a centrosome marker [68], and MKLP1, also referred to as KIF23, a marker for midbody remnants [69], a post-mitotic midbody-related structure thought to deliver material to the centrosome preceding cilia formation [64]. As shown in Figure 5, the 90413h antibody was found to label a single dot in approximately 1 out of 25–30 crypts, in agreement with the observed rarity of the primary cilium in the intestinal epithelium. 

As expected for polarized epithelial cells, the centrosomes were detected in the apical region of every cell while ARL13B stained with the 90413h antibody was generally detected under a single instance in the few crypts identified as being positive (Figure 6A, arrow). Interestingly, in most cases, ARL13B was detected adjacent to, but never superposed on, pericentrin-stained dots, as illustrated in Figure 6B–E and shown at a higher magnification in Figure 6E’. It is noteworthy that in the stroma, where most cell types exhibit a primary cilium, a similar pattern for ARL13B and pericentrin detection is observed (Figure 6F,G). 

The co-distribution of the primary cilium with the midbody remnants was also investigated. As reported in kidney polarized epithelial cells, it is hypothesized that only cells with a midbody remnant can become ciliated [64]. Using MKLP1 as a marker for midbody remnants [69], we investigated whether midbody remnants are detected in the intestinal crypt and their potential relation with the presence of the primary cilia in crypt intestinal cells. MKLP1-positive structures were found at a relatively low frequency in crypt cells. As shown in Figure 7 and Figure 8, MKLP1 staining was found in close juxtaposition with the primary cilium. For dual staining using the ARL13B 90413h and anti-MKLP1 antibodies, most of our observations identified the two structures side by side (Figure 7). Additionally, we note that MKLP1 was not regularly identified in the stroma. Similarly, the double staining of acetylated-α-tubulin with MKLP1 showed a similar co-distribution with the primary cilium and midbody remnants in positive crypts (Figure 8). 

The co-distribution of the ARL13b-positive structures with the +4 reserve stem cell marker BMI1 was then revisited using the ARL13b 90413h antibody. In contrast to the detection of ARL13b with the 17711-1-AP antibody, which appeared to cross-react with non-cilia structures under the immunofluorescence conditions used for tissue sections, the 90413h antibody detected only one cilium per positive crypt on average as mentioned above. While the odds of detecting a positive crypt displaying a primary cilium are low, the detection of a primary cilium at the luminal aspect of the same cell for which the nucleus is visible at the base in the same plane of the section must be even lower. Nevertheless, by examining a large number of tissue sections, we were able to confirm the previous observation that cells displaying a primary cilium are generally those expressing more intense levels of BMI1 (Figure 9). However, it is noteworthy that not all strongly BMI1-positive cells exhibited a primary cilium. In some cases, the primary cilium was seen on top of a single (Figure 9A,B) or a cluster of BMI1-positive cells (Figure 9C,D), but the majority of BMI1-positive crypt cells did not display an ARL13b-positive component. 

### 3.2. BMI1 and the Primary Cilia Are Detected by Human Intestinal Normal Crypt HIEC-6 Cells

The HIEC-6 cell line was generated to investigate human intestinal crypt functions and is the only normal crypt cell model currently available for this purpose [46,47]. Interestingly, HIEC-6 cells grow as undifferentiated cells in culture but can be induced toward a differentiated phenotype by the forced expression of pro-differentiation transcription factors [51] or toward LGR5+ primordial stem cells upon the activation of the WNT pathway [38]. Here, using normal non-stimulated cell culture conditions under which the HIEC-6 cells have no ability to differentiate [46,48], indirect immunofluorescence experiments have revealed that newly confluent cells constitutively expressed BMI1 nuclear staining while primary cilia were detected in most cells by anti-acetylated-α-tubulin (Figure 10D). BMI1 transcripts were also detected in HIEC-6 cells at all stages (Figure 10C).

### 3.3. The Primary Cilia and GLI1 Are Detected in Quiescent HIEC-6 Cells

Post-confluent HIEC-6 cells stop their expansion after five to ten days. However, HIEC-6 cells grow slowly and a significant proportion of the cells are not actively cycling, as illustrated by the detection of ciliated cells even before confluence. To better document the relation between cycling and the expression of the primary cilium, HIEC-6 cells were synchronized before they were investigated for primary cilium detection and BrdU incorporation. Under these conditions, the primary cilium was only detected in post-confluent cells using a combination of anti-ARL13b 17711-1-AP and anti-polyglutamylated tubulin antibodies (Figure 11A,B) at times when cell cycling is minimal, as evaluated by BrdU incorporation (Figure 11C). It is noteworthy that the 17711-1-AP antibody appeared to be specific for the primary cilium in these cells in contrast to the above observations on tissue sections in which non-cilia components were detected (as in Figure 1, Figure 2 and Figure 3). 

Quiescent HIEC-6 cells were also investigated for the expression of GLI1, the major mediator of the HH pathway. An indirect immunofluorescence analysis showed that most HIEC-6 cells display predominantly nuclear staining for immunoreactive GLI1 and primary cilia as detected by anti-acetylated-α-tubulin antibody (Figure 12A–D). Incidentally, GLI1 mRNA was detected in HIEC-6 at all stages but at much higher levels in post-confluent cells (Figure 12E).

### 3.4. The Primary Cilium Activates the Canonical HH Pathway in HIEC-6 Cells

To further investigate the involvement of the primary cilium in the regulation of the HH pathway in HIEC-6 cells, we have targeted the SMO receptor and GLI1 effector with pharmacological approaches.

In the first approach, confluent HIEC-6 cells were treated with 2 µM purmorphamine, an SMO agonist shown to activate the HH pathway [70]. We first analyzed the accumulation of full-length GLI3 at the tip of the primary cilia as an indicator of HH pathway activation [71]. As shown in Figure 13, a 24 h treatment with purmorphamine significantly increased the proportion of cilia stained with an anti-GLI3C antibody at their tips (Figure 13A–C). The expression of the HH downstream target genes *GLI1* and *PTCH1* was then investigated by qPCR and the expressions of both were found to have increased (Figure 13D,E). Variations in the *GLI1* and *PTCH1* levels were consistent with those observed between the various colorectal cancer cell lines [26].

In the second approach, confluent HIEC-6 cells were treated for 48 h with 5 µM GANT-61, an inhibitor of GLI-induced transcription [72]. The GANT-61 treatment inhibited the transcript expression of both *GLI1* and *PTCH1* in HIEC-6 cells (Figure 14A). Taken together, these observations indicate that the HH pathway is active in HIEC-6 cells and that the primary cilium is mediating its activity in intestinal crypt cells.

### 3.5. The HH Pathway Is Linked to a Quiescent State in HIEC-6 Cells

One interesting characteristic of post-confluent HIEC-6 cells is their ability to resume cell proliferation even after a prolonged quiescent period (Figure 14B). To verify the relation between the activity of the HH pathway and quiescence, 25- to 30-day-post-confluent HIEC-6 cells were treated for 48 h with GANT-61 or a control before being passed and allowed to recover for a 48 h period for BrdU staining. As shown in Figure 14C, the inhibition of GLI1 resulted in an acceleration in cell cycling, suggesting that the HH pathway regulates quiescence in human intestinal crypt cells.

## 4. Discussion

In this study, we discovered a small subset of crypt epithelial cells located just above the Paneth cells that bear a primary cilium, as deduced from immuno-labeling studies for the detection of primary cilia-associated components, such as acetylated-α-tubulin [59,60], polyglutamylated tubulin [65,66] and ARL13b [62,63]. In agreement with previous studies, all main cell types comprising the intestinal epithelium were found to be devoid of primary cilia [26,27,28,29,30]. This included most cells of the crypts that are cycling, such as primordial stem cells and transit-amplifying cells, as well as differentiated Paneth cells, another exocrine cell type that, like the acinar cell of the pancreas, does not exhibit primary cilia [73] and the terminally differentiated cells of the villus, such as goblet or absorptive cells. Incidentally, while most colorectal adenocarcinoma cell lines were found to display a primary cilium, the Caco-2 cell line, which can polarize and form a typical brush border similar to that of absorptive cells of the villus epithelium, was an exception [26], suggesting that luminal specialization may interfere with primary cilium formation in quiescent cells. The intestinal epithelium was thus considered to be an exception in comparison with most other tissues which display primary cilia, although primary cilia are present in intestinal tumors and adenocarcinoma cell lines [26,28,32,33]. The puzzling issue resulting from this observation is not the fact that the primary cilia are detected in intestinal cancer cells, considering the growing indications that the primary cilia are functionally involved in a variety of other neoplasms [1,74], but the expression of primary cilia in tumor cells that lack them in their normal counterparts [25]. While it is too early to speculate about a possible relation between the primary cilia in intestinal adenocarcinoma cells and our discovery of a small subset of ciliated cells in the normal intestinal crypt which may be considered stem cells, this latter observation at least provides the first piece of evidence that some normal intestinal cells can express a primary cilium under specific circumstances. 

The fact that cells bearing a primary cilium were predominantly observed near the bottom of the crypt but always above the Paneth cells is noteworthy. In mouse, this cell location corresponds to the position of +4 label retained cells described several years ago by Potten et al., as reviewed in [75] and further characterized as BMI-1 expressing cells able to restore the intestinal epithelium after the loss of the primordial LGR5+ stem cell population [35,36]. Further studies have confirmed the concept that rare damage-resistant and quiescent +4 reserve intestinal stem cells can re-activate the primordial stem cell pool and facilitate regeneration as summarized by Bankaitis et al. [40]. One important issue that remains about these +4 reserve stem cells is their identification. Several markers have been reported to identify the +4 reserve stem cells in mouse, including BMI1, leucin-rich repeat and immunoglobulin-like domain-1 (LRIG1), HOP homeobox (HOPX) and mouse telomerase reverse transcriptase (MTERT); additionally, lineage-tracing studies have confirmed that cells expressing them behave as reserve stem cells [40,41]. However, the absolute specificity of these markers for the +4 reserve stem cells has been criticized based on the broader expression of their transcripts and some lineage-tracing experiments while there is more and more evidence that many quiescent cell types are present in the intestinal epithelium, such as some secretory progenitors [76] which also participate in regenerative responses following injury [40,41]. In the light of this complexity, it is interesting to note that the primary cilium appeared to be restricted to rare cells corresponding to quiescent +4 reserve stem cells of the human intestinal crypt. Furthermore, the adjacent location of the primary cilium with midbody remnants in these rare cells supports the model proposed by Labat-de-Hoz et al., in which only cells with a midbody remnant can become ciliated [64]. Co-staining for the detection of the primary cilium and nuclear BMI1 identified corresponding cells or cell clusters, suggesting that at least some of the BMI1 expressing cells are related to the quiescent +4 reserve stem cells in the human intestinal crypt, although it is clear that not all BMI1-positive cells display a primary cilium. It is noteworthy that the identification of BMI1 as a marker for the +4 reserve stem cells was obtained through various strategies such as the use of a reported gene expression system in experimental animal models [35,42,77], while in human models, the BMI1 protein was mainly detected in gastrointestinal cancer cells in which it is overexpressed but not or only weakly in their normal counterparts [78,79,80]. In this study, the immunodetection of the BMI1 protein in the human intestine showed discrete cells or cell clusters in the lower crypt region that were stained more intensively than their surroundings showing weak but consistent positive staining, an observation in agreement with the fact that BMI1, while mainly expressed by +4 reserve stem cells, also displays a broader expression pattern in other intestinal epithelial cells [77,81,82]. 

The ability of human intestinal stem cells to produce a functional primary cilium when maintained under a quiescence state was further investigated using the HIEC-6 cell line. Being normal and non-transformed, HIEC-6 cells exhibit typical crypt cell proliferative and undifferentiated characteristics and have been proven to be useful for studying human crypt cell functions, such as proliferation, cell survival, cell–matrix interactions, metabolism and the inflammatory response [46,48]. Interestingly, the ectopic expression of pro-differentiation factors, such as CDX2 and HNF1α, and the abolition of polycomb repressive complex 2 epigenetic regulation [51,83] demonstrated that HIEC-6 cells maintain the ability to undertake a differentiation program. It is also pertinent to note that normal HIEC-6 expresses a variety of intestinal stem/progenitor cell markers, such as *BMI1*, doublecortin-like kinase 1 (*DCAMKL1*), Mushashi-1 (*MSI1*), epithelial cell adhesion molecule (*EPCAM*), and *CD44,* while expressing very low levels of *LGR5* and WNT/β-catenin activity, suggesting that HIEC-6 cells behave as reserve stem cells. Interestingly, the activation of the WNT pathway by normal ligands in HIEC-6 triggered a robust expression of primordial stem cell markers, such as LGR5 and pleckstrin homology-like domain family member-1 (PHLDA1) at both the transcript and protein levels, confirming the potential of HIEC-6 cells to acquire a primordial stem cell phenotype [38]. Another interesting feature that was revealed in the present study is the ability of HIEC-6 cells to fully, but reversibly, stop proliferation when maintained at confluence for a prolonged period and express a primary cilium. 

The accumulation of GLI1 at both the transcript and protein levels in ciliated HIEC-6 cells at confluence suggest that the HH pathway is active at quiescence [74], implying an autocrine mechanism. To confirm this possibility, we stimulated the HH pathway with purmorphamine, an SMO agonist [70] and found a further activation of the HH pathway, as evaluated by an increase in the accumulation of full-length GLI3 at the tip of the primary cilia and in the expression of the two downstream targets genes, *GLI1* and *PTCH1.* From these data, it appears that HIEC-6 cells spontaneously display a primary cilium when they become quiescent at confluence and that this primary cilium mediates the HH signals. 

Quiescence, likely resulting from a contact inhibition-related mechanism, allows primary cilium assembly since the basal body and associated components are no longer required for the mitotic spindle [24,25]. We investigated whether the resulting autocrine HH signaling in HIEC-6 is simply an outcome of the presence of a functional primary cilium acquired at quiescence or it also contributes to cell cycle arrest. For this, we exploited the characteristic of hyper-confluent HIEC-6 cells in which the cell cycle is completely stopped but normal cell proliferation kinetics can be restored after a few passages. For these experiments, proliferation was assessed 48 h after the first passage to evaluate the effects of GLI1 accumulation in confluent cells. The pharmacological inhibition of the transcriptional activity of GLI accelerated the restoration of normal proliferation kinetics, suggesting that the HH pathway contributes to the inhibition of the cell cycle in quiescent human intestinal crypt cells. There are several target genes susceptible to having their expression regulated by GLI transcription factors, some being pro-proliferative, others anti-proliferative, being tissue/cell type dependent [4,74,84,85]. Future studies will be needed to characterize the specific events downstream from the HH pathway that modulate quiescence in human intestinal reserve stem cells.

## 5. Conclusions

In this study, as summarized in Figure 15, we have discovered that primary cilia are present in a subset of cells located in the lower part of the crypts, just above the Paneth cell population, a position that coincides with the BMI1-positive quiescent +4 reserve stem cells reported in the intestinal crypt. Using HIEC-6 cells as a normal epithelial proliferative crypt cell model, we showed that primary cilia assembly correlates with a quiescent state and that this primary cilium mediates HH signals which in turn appear to contribute to the inhibition of the cell cycle. These results provide further evidence of the existence of a quiescent cell population in the human small intestine and suggest the potential for new modes of regulation in stem cell dynamics.

## Figures and Tables

**Figure 1 cells-12-01059-f001:**
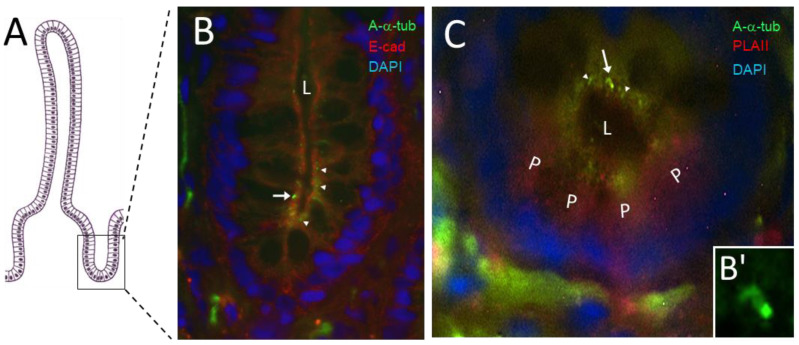
Primary cilia in the human small intestine. (**A**) Schematic of the crypt–villus axis showing the crypt region depicted in B and C. (**B**) Representative immunodetection of primary cilia using anti-acetylated-α-tubulin (A-α-tub; arrow, green staining) in the epithelial cells (E-cadherin, E-cad; red staining) of a crypt in the human adult small intestine (L, lumen). Nuclei were stained with DAPI (blue staining). Smaller structures that could be related to either midbody remnants [64] or centrosomes (see below) were also detected (arrowheads). (**C**) 3D reconstruction of double immunostaining of anti-α-acetylated-tubulin (green), showing primary cilia (arrow) and smaller structures (arrowheads) as well as anti-phospholipase A2 (red), a Paneth cell marker (L, lumen; P, Paneth cell). Nuclei were stained with DAPI (blue) and are out of focus because they are present on a lower plane of focus. Scale bars are equal to 10 µm. (**B’**) Higher magnification of the primary cilium seen in B (arrow).

**Figure 2 cells-12-01059-f002:**
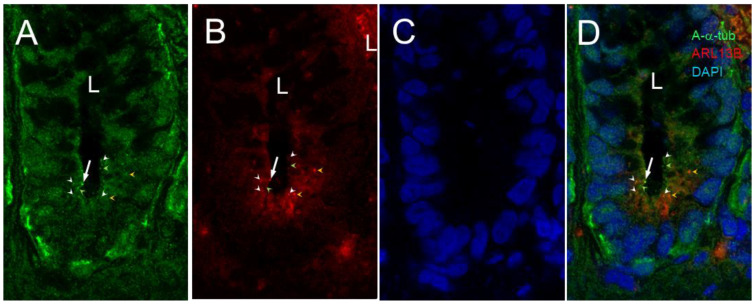
Primary cilia in the lower half of the crypts of the human small intestine. Representative confocal imaging for the detection of primary cilia using anti-acetylated-α-tubulin (A-α-tub; green staining, arrow in (**A**,**D**)) and anti-ARL13B antibody (red staining, arrow in (**B**,**D**)) in epithelial cells of the human adult small intestine (L, lumen of the crypt). Nuclei were stained with DAPI (blue) (**C**,**D**). Note that anti-acetylated-α-tubulin and anti-ARL13B stained smaller dots that were co-stained (white arrowheads) or not stained (green and orange arrowheads) in (**A**,**B**,**D**). Scale bar is equal to 10 µm.

**Figure 3 cells-12-01059-f003:**
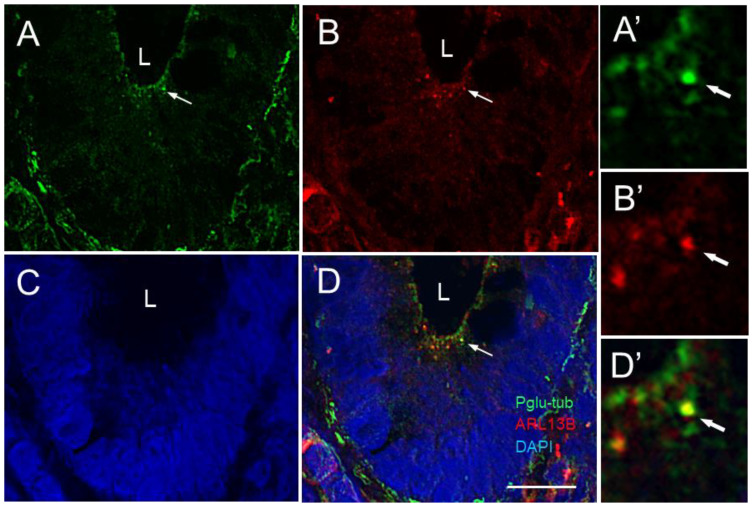
Primary cilia in the lower part of the crypts of the human small intestine. Representative confocal imaging for the detection of primary cilia using anti-glutamylated tubulin (Pglu-tub; green staining, arrow in (**A**,**D**)) and anti-ARL13B antibody (red staining, arrow in (**B**,**D**)) in epithelial cells of the human adult small intestine (L, lumen of the crypt). (**A’**,**B’**,**D’**) are higher magnifications of the (**A**,**B**,**D**) panels. Nuclei were stained with DAPI (**C**,**D**). As observed above, the 17711-1-AP antibody (**B**,**D**) identified dots that were only weakly stained or not stained with the anti-Pglu-tub antibody. Scale bar is equal to 10 µm.

**Figure 4 cells-12-01059-f004:**
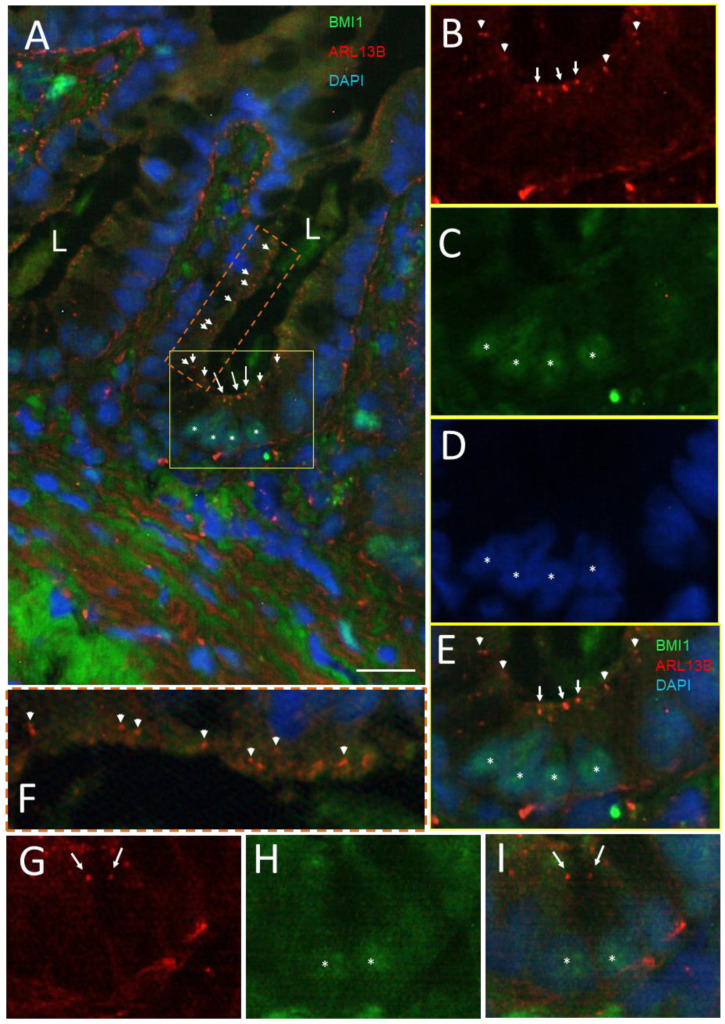
Primary cilia in BMI1-positive cells of the crypts of the human small intestine. (**A**) Representative immunodetection of primary cilia using the anti-ARL13b 17711-1-AP antibody (arrow, red staining) and positive BMI1 nuclei (green staining, stars) in epithelial crypt cells of the human adult small intestine (L, lumen of the crypts). Nuclei were stained with DAPI (blue). The square delimits the portion of the lower crypt shown in panels (**B**–**E**) and the rectangle delimits the portion of the luminal crypt epithelium shown in panel (**F**). (**B**–**E**) Higher magnification of the lower crypt region showing a cluster of positive 17711-1-AP structures in the luminal aspect of lower crypt cells (arrows in **B**,**E**) and positive nuclei for BMI1 staining (stars in **C**,**E**). Arrowheads in the (**A**,**B**,**E**,**F**) panels identify smaller 17711-1-AP-stained dots that do not seem to be related to the primary cilium. (**G**,**H**) Another region showing the expression of 17711-1-AP-stained structures (arrows) in cells displaying positive BMI-1-stained nuclei (stars). Nuclei were stained with DAPI (**A**,**D**,**E**,**I**). Scale bar is equal to 10 µm.

**Figure 5 cells-12-01059-f005:**
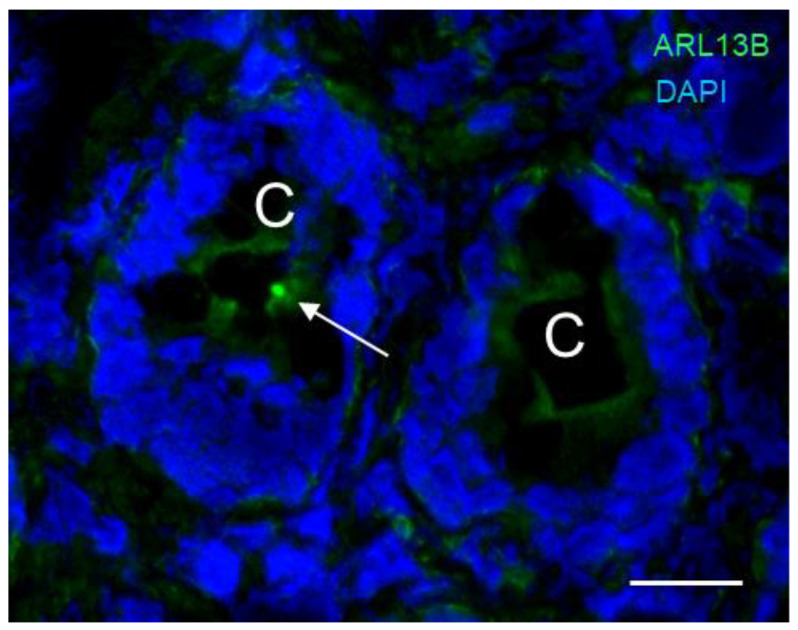
Primary cilia in the lower third of a crypt of the human small intestine. A, Representative immunodetection of primary cilia using the anti-ARL13b 90413h antibody (green staining) (arrow) in one of the crypts (C). Nuclei were stained with DAPI (blue staining). Scale bar is equal to 10 µm.

**Figure 6 cells-12-01059-f006:**
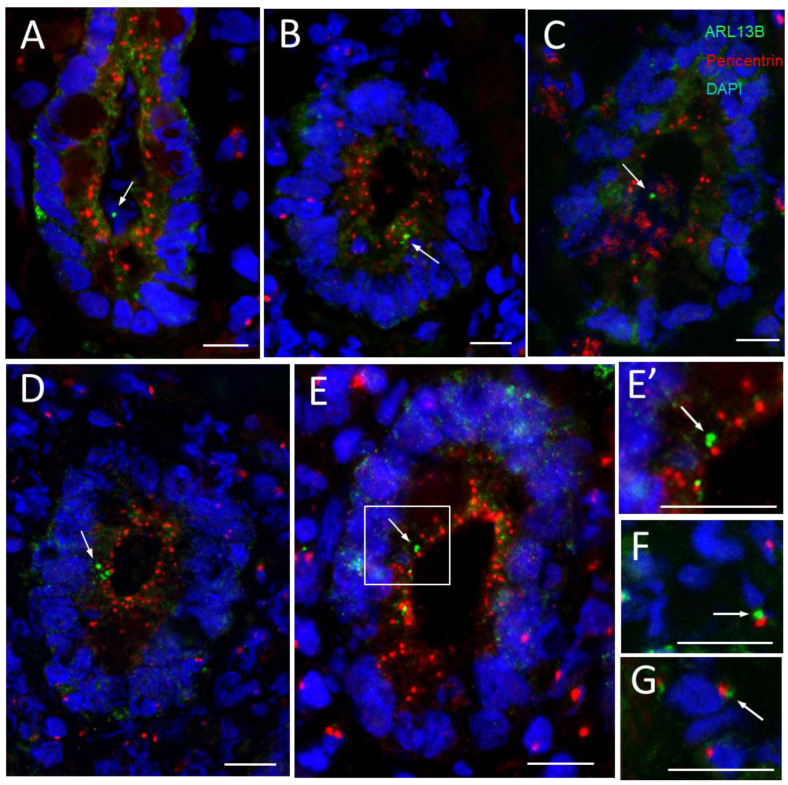
Primary cilia and centrosomes in the lower third of the crypts of the human small intestine. (**A**–**E**) Representative immunodetection of primary cilia using the anti-ARL13b 90413h antibody (green staining, arrow) and of centrosomes using an anti-pericentrin antibody (red staining) in various crypts. In most cases of positive crypts, the 90413h antibody stained one predominant dot (**A**–**E**) that was localized adjacent to a positive pericentrin-labeled dot, as in (**B**–**E**). (**E’**) Higher magnification of the section delimited by the square in (**E**) showing adjacent dots stained by ARL13B and pericentrin. (**F**,**G**) are high magnifications of the stroma around the crypts showing a similar distribution of ARL13B and pericentrin in ciliated fibroblasts. Nuclei were stained with DAPI (blue staining). Scale bars are equal to 10 µm.

**Figure 7 cells-12-01059-f007:**
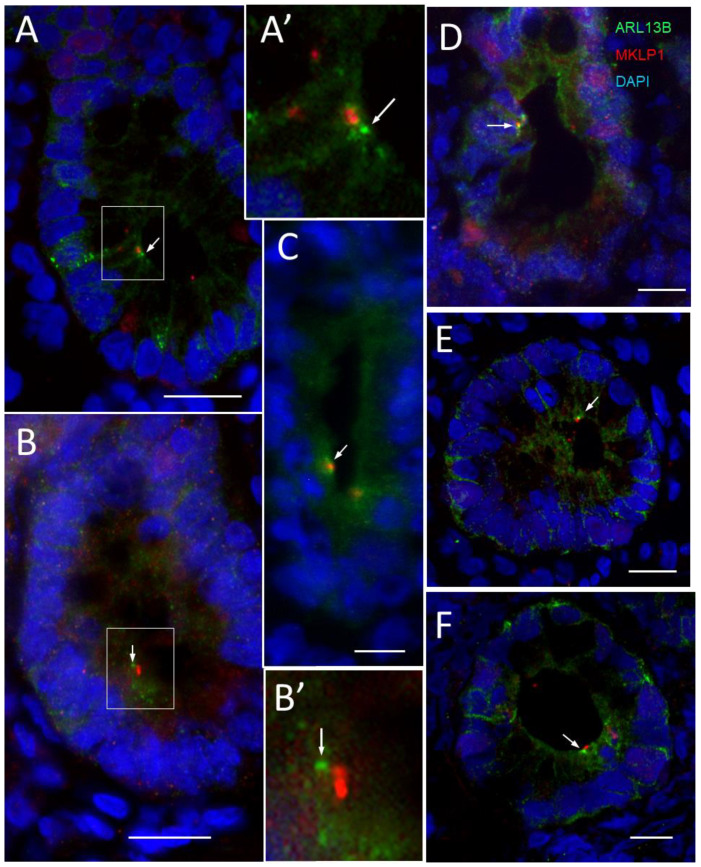
Primary cilia and midbody remnants in the lower third of the crypts of the human small intestine. (**A**–**E**) Representative immunodetection of primary cilia using anti-ARL13b 90413h antibody (green staining, arrow) and of midbody remnants using an anti-MKLP1 antibody (red staining) in various crypts. In most cases of positive crypts, the 90413h antibody stained one predominant dot (**A**–**F**) that was localized adjacent to a positive MKLP1-labeled dot. (**A’**,**B’**) Higher magnification of the section delimited by the squares in (**A**,**B**) showing adjacent dots stained by ARL13B and MKLP1. Nuclei were stained with DAPI (blue staining). Scale bars are equal to 10 µm.

**Figure 8 cells-12-01059-f008:**
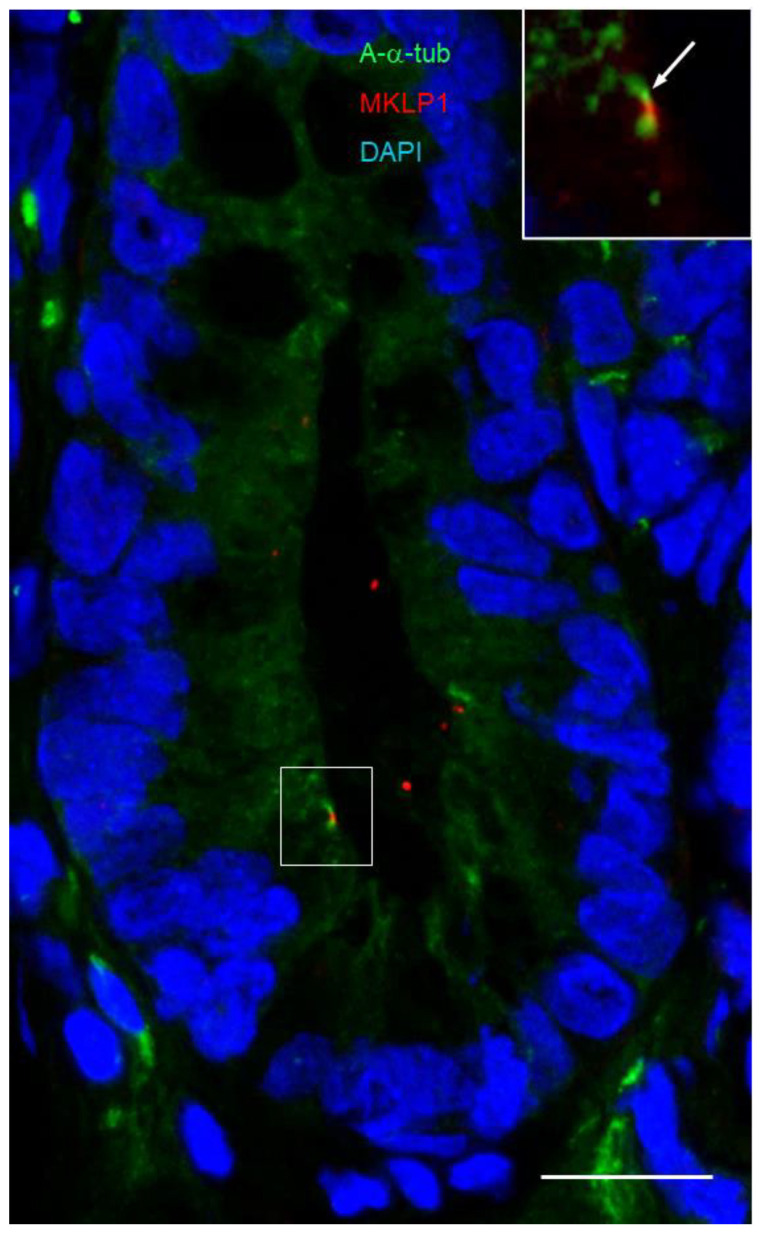
Primary cilia and midbody remnants in the lower third of the crypts of the human small intestine. Representative immunodetection of primary cilia using anti-acetylated tubulin antibody (green staining) and midbody remnants using an anti-MKLP1 antibody (red staining) in the lower portion of a crypt. The 90413h antibody stained one predominant structure that was localized adjacent to a positive MKLP1-labeled dot. Insert in the upper right corner, higher magnification of the section delimited by the square showing the co-staining. Nuclei were stained with DAPI (blue staining). Scale bar is equal to 10 µm.

**Figure 9 cells-12-01059-f009:**
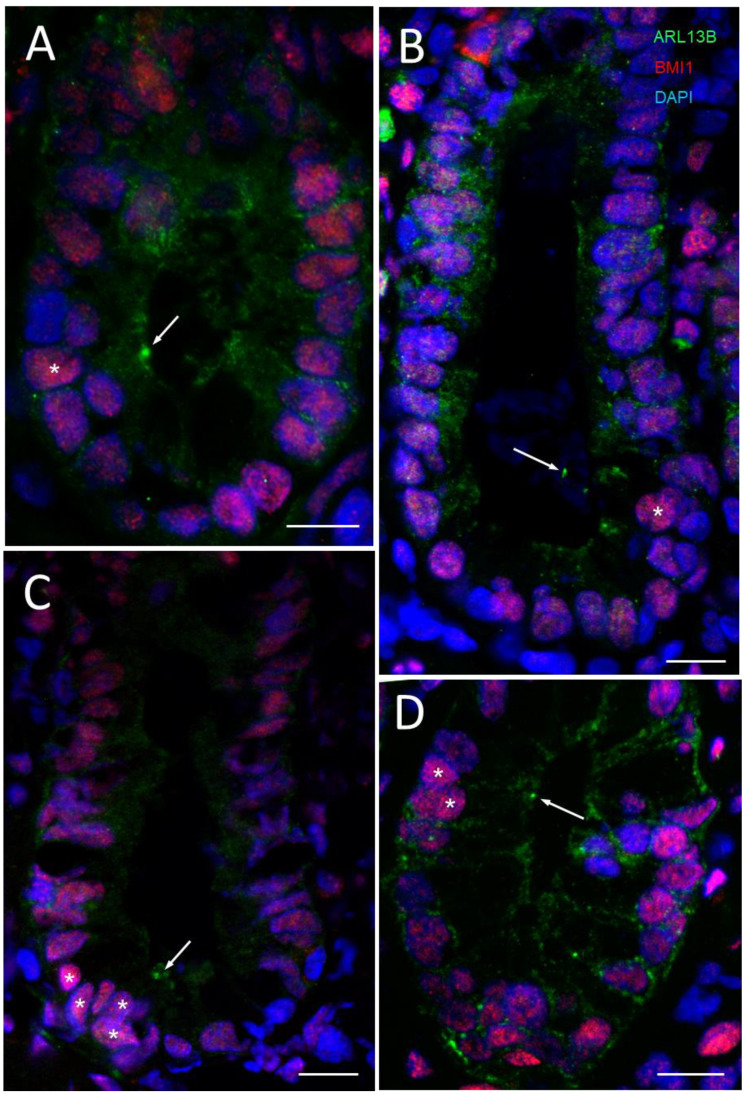
Primary cilia in association with BMI1-positive cells of the crypts of the human small intestine. (**A**) Representative immunodetection of primary cilia using the anti-ARL13b 90413h antibody (arrows, green staining) and positive BMI1 nuclei (red staining, stars) in epithelial crypt cells of the human adult small intestine. In some instances, single nuclei staining was observed to be more intensive for BMI1 in relation to the detection of cilia (**A**,**B**) while in other cases, a cluster of BMI1-positive nuclei coinciding with ARL13b-positive structures was noted (**C**,**D**). Nuclei were stained with DAPI (blue). Scale bars are equal to 10 µm.

**Figure 10 cells-12-01059-f010:**
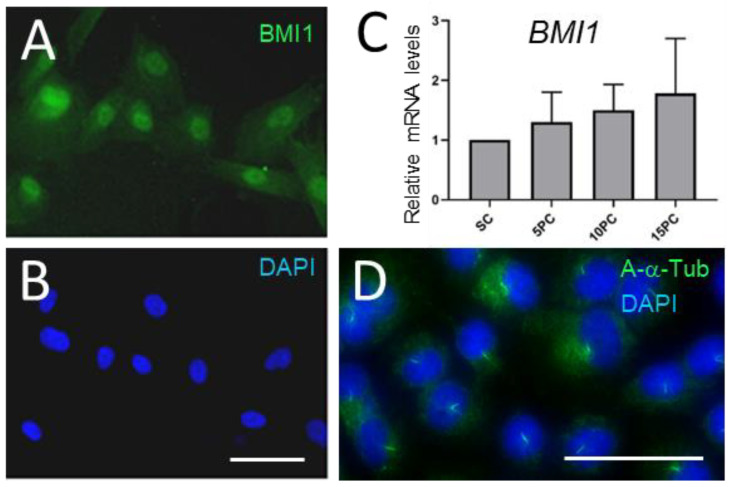
Constitutive expression of BMI1 and assembly of the primary cilia in HIEC-6 cells. (**A**,**B**): Representative immunodetection of BMI1 (**A**) and DAPI nuclear co-staining (**B**) in newly confluent HIEC cells. (**C**): BMI1 transcript expression in subconfluent (SC) and 5-, 10- and 15-day post-confluent (5PC, 10PC and 15PC) HIEC-6 cells. (**D**): Immunodetection of acetylated-α-tubulin (A-α-tub) in newly confluent HIEC-6 (green) cells. Scale bars are equal to 25 µm.

**Figure 11 cells-12-01059-f011:**
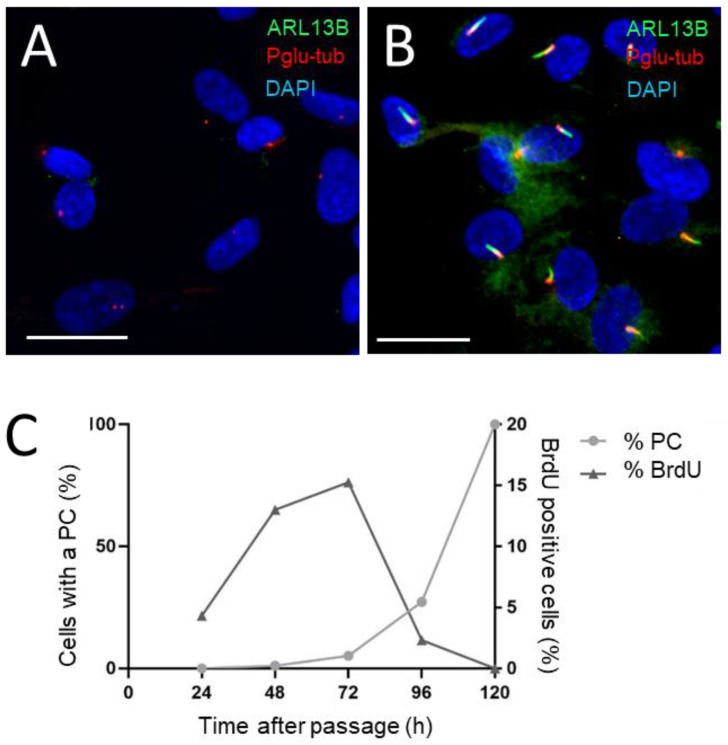
Expression of the primary cilia in quiescent HIEC cells. (**A**,**B**): Representative indirect immunofluorescence for the detection of ARL13b with 17711-1-AP (green staining) and polyglutamylated tubulin (Pglu-tub; red) in subconfluent (**A**) and postconfluent (**B**) HIEC cells. (**C**). Representative experiment showing primary cilium (PC) and BrdU counts in percentage of total DAPI stained cells in synchronized cells at 24 h intervals after passage. PC and BrdU counts were performed in separate dishes. The experiment was repeated three times. Bars are equal to 10 µm.

**Figure 12 cells-12-01059-f012:**
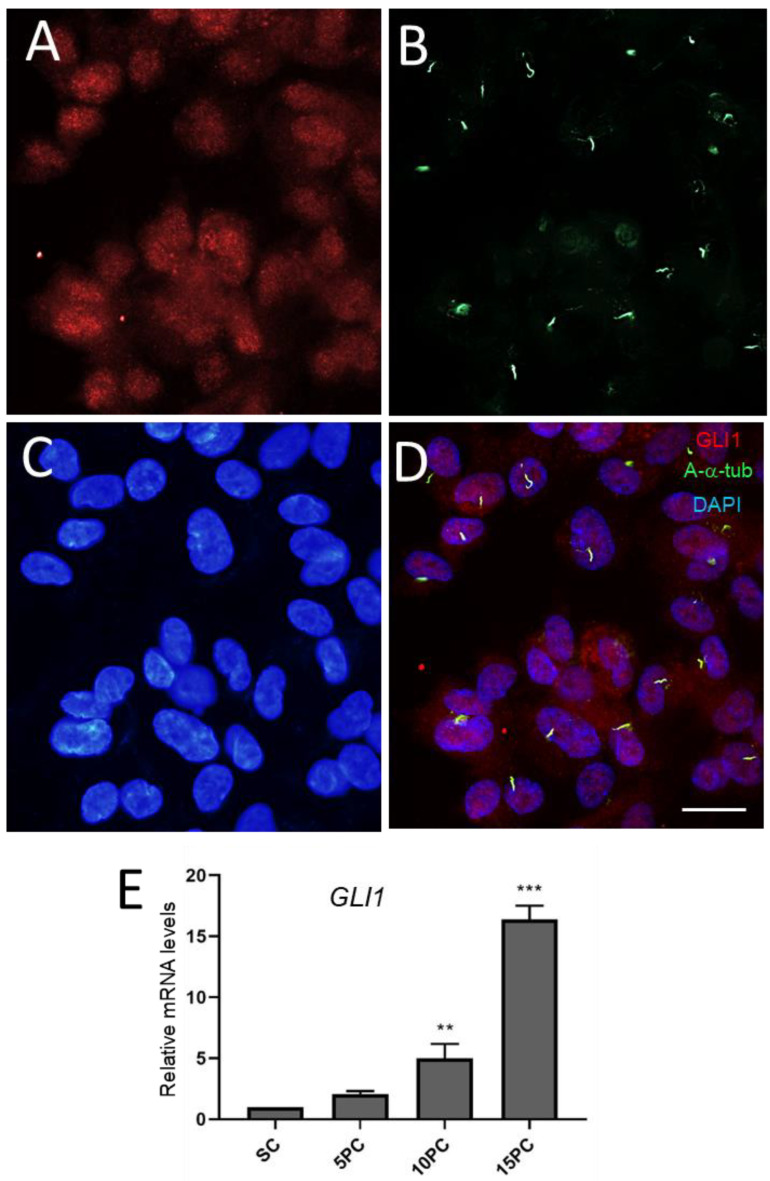
Accumulation of GLI1 expression in quiescent HIEC cells. (**A**–**D**): Representative indirect immunofluorescence for the detection of GLI1 (**A**,**D**, red staining) and acetylated-α-tubulin (A-α-tub; **B**,**D**, green staining) in post-confluent HIEC cells. Nuclei were stained with DAPI (**C**,**D**). Scale bar is equal to 10 µm. (**E**): *GLI1* transcript expression in subconfluent (SC) and 5-, 10- and 15-day post-confluent (5PC, 10PC and 15PC) HIEC-6 cells. **, *p* < 0.01; ***, *p* < 0.0005.

**Figure 13 cells-12-01059-f013:**
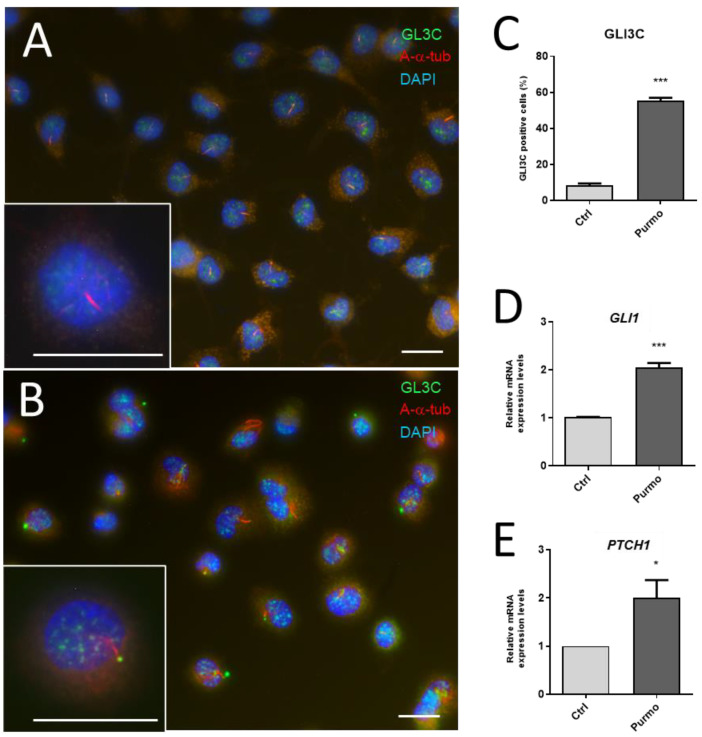
Stimulation of the HH pathway in response to purmorphamine in HIEC cells. Quiescent post-confluent HIEC cells were untreated (**A**) or treated with 2µM purmorphamine (Purmo) (**B**) for 24 h and analyzed for the HH activity marker GLI3C (**A**–**C**) and expression of downstream target genes *GLI1* (**D**) and *PTCH1* (**E**). Primary cilia were detected using an anti-acetylated-α-tubulin (A-α-tub; red staining) in most post-confluent HIEC cells (**A**,**B**). Accumulation of GLI3C expression at the tip of primary cilia with an anti-GLI3C (green staining) was found in ~10% of control cells (**A**,**C**, Ctrl) while it was detected in more than 50% of the Purmo-treated cells (**B**,**C**, Purmo). Bars in (**A**,**B**) are equal to 10 µm. (**D**,**E**): Expression of *GLI1* and *PTC1* transcripts was also significantly increased in Purmo-treated cells. *, *p* < 0.05; ***, *p* < 0.001.

**Figure 14 cells-12-01059-f014:**
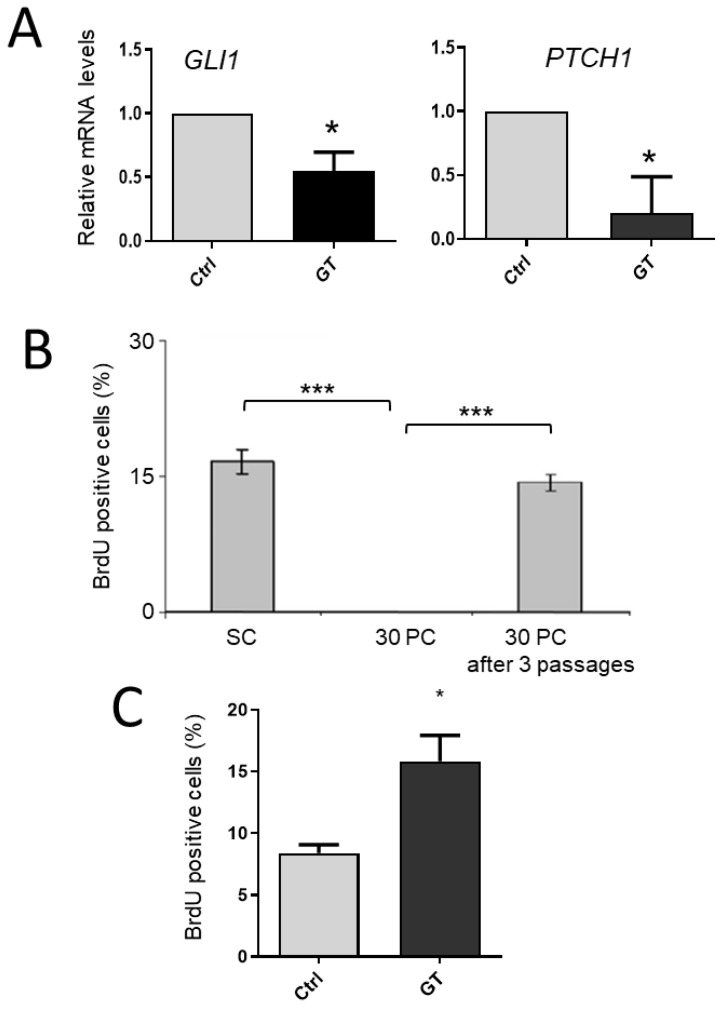
Quiescence of post-confluent HIEC cells is reversible and is regulated by the HH pathway. (**A**): In post-confluent HIEC-6 cells, the GLI1 inhibitor GANT-61 (GT) reduced the expression of both GLI1 and PTCH1 transcripts, the main HH downstream target genes, after 48 h treatment. (**B**): In contrast to sub-confluent (SC) cells, HIEC-6 cells that maintained up to 30 days of post-confluent (30PC) culture did not synthesize DNA as evaluated by the lack of BrdU staining. However, BrdU staining was restored to basal levels after 3 passages. (**C**): 20+ day-post-confluent HIEC-6 cells were treated for 48 h with GT before being passed and allowed to recover for 48 h then processed for BrdU staining and cell counting. *, *p* < 0.05; ***, *p* < 0.001.

**Figure 15 cells-12-01059-f015:**
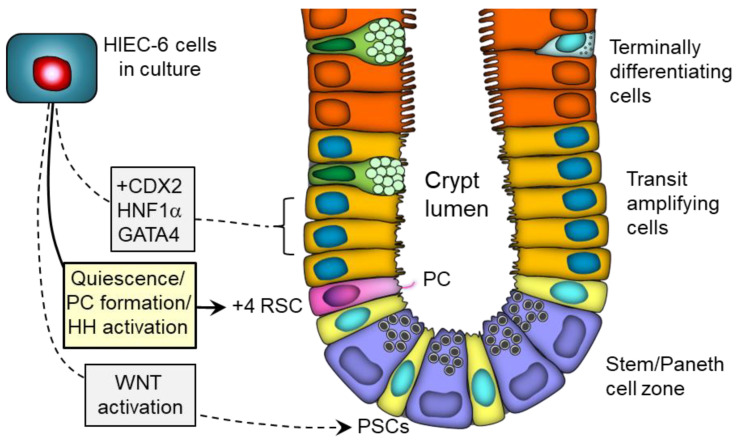
Schematic illustration of the findings of the current study. Primary cilium (PC)-bearing cells (pink) were found in the lower crypt just above the stem/Paneth cell zone at a frequency estimated at one cell per crypt in the human small intestine and corresponding to the +4 reserve stem cells (RSC) described previously. HIEC-6 cells were then used to further investigate the expression of the PC and signaling. HIEC-6 cells were previously found to be unique in their abilities to undertake a differentiation program under the influence of pro-differentiation transcription factors, such as CDX2 and HNF1α [51,83], as well as to be induced to adopt a primordial stem cell (PSC) phenotype upon activation of the WNT pathway [38]. Herein, we showed that post-confluent HIEC-6 cells express a PC when becoming quiescent and that this PC mediates the activation of the HH pathway, which in turn appears to regulate the cell cycle.

## Data Availability

Not applicable.

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
