# Peer review of "Primary Cilium Identifies a Quiescent Cell Population in the Human Intestinal Crypt"

_cells, 2023, doi:10.3390/cells12071059_

Round 1

Reviewer 1 Report

In this manuscript, Senicourt at al attempt to show that primary cilia are detected in a subset of cells located deep in the 0 crypts slightly above the Paneth cell population and could correspond to BMI1 positive cells that are mostly quiescent, resistant to stress and have the capacity to regenerate LGR5 positive cells after tissue injury. The rationale is very interesting and could impact understanding of villi formation/function in different conditions. However, major points need to be addressed before publication.

Search for primary cilia in the reserve stem cells of the human intestinal crypt section

figures are complex to analyze.

- Fig1 is unclear. Fig.2 ARL13b staining is not obvious either. Can authors use antibodies against acetylated alpha tubulin and pericentrin (or gamma tubulin)? It would allow visualization of both cilia (in reduced number of cells) and centrosome (in all cells) more easily. A schematic of the sections could also help to visualize villi.

-In figure 1, authors should also use other paneth cell markers such as lysozyme that give a stronger signal

- In Fig3, BMI1 staining seems also very weak. Why a fraction only of BMI1 cells  have a cilia? What is the rationale behind this observation?

- In this section, authors should give more details on human intestinal samples collected: age of patients, sex, disorders of patients? It could be interesting to correlate results obtained to different conditions, if known.

“BMI1 and the primary cilia are expressed by human intestinal normal crypt HIEC-6 cells” section

- In Fig4, authors should present double staining rather than separate stainings.  Fig5 is convicing. Maybe Fig4 and 5 could be compiled to strengthen the data.

- Rationale of Fig6 should be explained more clearly. Why looking at Gli1 only? What is the localization of other well known ciliated expressed proteins? (Gli1 staining is not convincing to me).

 "The primary cilium activates the canonical HH pathway in HIEC-6 cells" section

- In a first approach, confluent HIEC-6 cells were treated with 2 μM purmorphamine and authors look at accumulation of full length GLI3 at the tip of the primary cilia. What is the localization of other effectors? Can their activation be looked at using western blot?  It could help quantify GLI3FL:GLI3R ratio for example.

-  Do the cells respond to other Hh Signaling Transduction activation molecules? Authors could stimulate cells by the addition of recombinant human SHH (rhSHH) or smoothened agonist (SAG) followed by quantification of cilia components by immuno or by analyzing the expression of two Hh target genes, such as GLI1 and PTCH1.

Author Response

Thank you for your positive remarks and suggestions.

Remark:

- Fig1 is unclear. Fig.2 ARL13b staining is not obvious either. Can authors use antibodies against acetylated alpha tubulin and pericentrin (or gamma tubulin)? It would allow visualization of both cilia (in reduced number of cells) and centrosome (in all cells) more easily. A schematic of the sections could also help to visualize villi.

Response:

The difficulty with the identification of the primary cilium in the intestine is its rarity. As mentioned in the text, on average, only one cilium was detected for each 15 crypts examined. Furthermore, when tested in preliminary experiments, the high intensity of the staining for gamma-tubulin complicated the detection of acetylated alpha tubulin because of the likely presence of midbody remnants as suggested by reviewer 2. As suggested, a schematic of the section was added to help localize the region of the crypt-villus axis depicted in Fig 1 (and 2).   

Remark:

-In figure 1, authors should also use other paneth cell markers such as lysozyme that give a stronger signal.

Response:

We tested various markers to identify Paneth cells such as lysozyme and defensin 5 but both provided relatively intense luminal staining and only stained the most luminal part of the Paneth cells while PLA2 stained the Paneth cells uniformly without luminal background. 

Remark:

- In Fig3, BMI1 staining seems also very weak. Why a fraction only of BMI1 cells have a cilia? What is the rationale behind this observation?

Response:

BMI1 detection in cryosections of the normal human intestine is notoriously difficult as compared to colorectal cancer sections or via lineage tracing experiments with genetic animal models. The rationale behind the fact that not all BMI1 positive cells exhibit a primary cilium is not clear but was mentioned in the revised version as: " (...) the finding that at least some of the BMI1 expressing cells are related to the quiescent +4 reserve stem cells in the human intestinal crypt although it is noteworthy that not all BMI1 positive cells display a primary cilium." lines 401-403.       

Remark:

- In this section, authors should give more details on human intestinal samples collected: age of patients, sex, disorders of patients? It could be interesting to correlate results obtained to different conditions, if known.

Response:

All samples were obtained from normal donors. As suggested, age and sex information was provided: "The average age of the donors was 49.5 ± 21.0 (21-82) and were 46% female and 54% male.", lines 93-94.

“BMI1 and the primary cilia are expressed by human intestinal normal crypt HIEC-6 cells” section

Remark:

- In Fig4, authors should present double staining rather than separate stainings.  Fig5 is convicing. Maybe Fig4 and 5 could be compiled to strengthen the data.

Response:

Unfortunately, these preliminary experiments were performed separately but since a vast majority of the cells were positive for both BMI1 and acetylated-a-tubulin, double staining would not add much.  For more clarity, we would like to keep Fig. 4 and 5 separate.

Remark:

- Rationale of Fig6 should be explained more clearly. Why looking at Gli1 only? What is the localization of other well known ciliated expressed proteins? (Gli1 staining is not convincing to me).

Response:

Nuclear Gli1 expression was used as a first indicator of HH activity in confluent ciliated HIEC-6 cells. Indeed, as observed by qPCR, transcript levels of GLI1 were significantly increased in post-confluent/quiescent HIEC-6 cells and as mentioned in the discussion, a potential indication that HH activity may act under an autocrine mechanism (lines 423-425).   

 "The primary cilium activates the canonical HH pathway in HIEC-6 cells" section

Remarks:

- In a first approach, confluent HIEC-6 cells were treated with 2 μM purmorphamine and authors look at accumulation of full length GLI3 at the tip of the primary cilia. What is the localization of other effectors? Can their activation be looked at using western blot?  It could help quantify GLI3FL:GLI3R ratio for example.

-  Do the cells respond to other Hh Signaling Transduction activation molecules? Authors could stimulate cells by the addition of recombinant human SHH (rhSHH) or smoothened agonist (SAG) followed by quantification of cilia components by immuno or by analyzing the expression of two Hh target genes, such as GLI1 and PTCH1.

Response:

The purpose of these experiments was to evaluate whether the primary cilia could be involved in autocrine activation of the HH pathway in intestinal cells. While the localization and expression of other effectors could be tested, the accumulation of full length GLI3 and increase in GLI1 and PTCH1 expression in response to the SMO agonist and the inhibition of the latter two with the GLI1 inhibitor suggests that this is the case. Although interesting these additional experiments would be beyond the aim of the study. Also please note that the 10-days allowed by the Editor for sending the revised manuscript would not give us time to perform these experiments.

Thank you,  

Reviewer 2 Report

In this study, Sénicourt et al analyze the existence of primary cilia in the crypts of the human small intestine. They find that all main cell types in the intestinal epithelium —including most cells of the crypts, such as primordial stem cells and Paneth cells, as well as the terminally differentiated cells of the villus, such as goblet or absorptive cells— are devoid of primary cilia. Embedded in this background of unciliated cells, they authors identify a small subset of cells located above the Paneth cells, which the authors characterize as BMI1-positive cells, with a primary cilium (Figs. 1-3). For the rest of the experiments, they use HIEC-6 cells, a human intestinal cell, as a crypt cell model. They show that quiescent HIEC-6 cells are positive for the BMI1 marker and that have a genuine primary cilium —with elongated morphology and positive for Arl13b and polyglutamylated tubulin— that is functional in activation of the Hedgehog signaling pathway (Figs. 4-8).

The article is interesting and well written, but there are some crucial concerns about the identification of primary cilia in Figs. 1-3 that must be addressed.

Major concerns:

- Fig. 1 and  Fig 2: It is not clear that the structure stained with anti-acetylated tubulin (arrow) is a primary cilium, since it does not have the typical elongated structure of cilia and does not protrude into the lumen. The authors claim that the other structures labeled with the antibody (arrowheads) are centrosomes. However, centrosomes do not typically stain for acetylated tubulin unless the cells are subjected to specific treatments prior to staining. The structures appear to be midbody remnants, which are known to contain ciliary markers, rather than primary cilia or centrosomes. To rule out that this possibility, staining experiments with antibodies to midbody remnant markers, such MKLP1 or PRC1, that do not label primary cilia or centrosomes should be performed. Further characterization of the structures using electron microscopy would clarify the presence of primary cilia in the crypt cells.

Fig. 3: Again, the staining with a ciliary marker (in this case Arl13b) shows dotted rather than elongated structures. The structures appear to be located apically, near the tight junctions, a location that is typical for midbody remnants. As in Figs. 1 and 2, further characterization of these structures is needed.

Author Response

Thank you for your positive remarks and suggestions.

Remarks:

The article is interesting and well written, but there are some crucial concerns about the identification of primary cilia in Figs. 1-3 that must be addressed.

- Fig. 1 and  Fig 2: It is not clear that the structure stained with anti-acetylated tubulin (arrow) is a primary cilium, since it does not have the typical elongated structure of cilia and does not protrude into the lumen. The authors claim that the other structures labeled with the antibody (arrowheads) are centrosomes. However, centrosomes do not typically stain for acetylated tubulin unless the cells are subjected to specific treatments prior to staining. The structures appear to be midbody remnants, which are known to contain ciliary markers, rather than primary cilia or centrosomes. To rule out that this possibility, staining experiments with antibodies to midbody remnant markers, such MKLP1 or PRC1, that do not label primary cilia or centrosomes should be performed. Further characterization of the structures using electron microscopy would clarify the presence of primary cilia in the crypt cells.

Response:

The difficulty with the identification of the primary cilium in the intestine is its rarity. As mentioned in the text, on average, only one cilium was detected for each 15 crypts examined. Even in colorectal tumours where primary cilia are observed more frequently (please see ref. 28), the typical elongated structure seen in vitro, on colorectal cancer cells or as here with HIEC-6 cells, was only seen on a few occasions as shown in Fig 1C, using reconstituted deconvoluted images. In other tissues such as the nephron, where a large proportion of the epithelial cells display primary cilia, it is possible to distinguish a few of these elongated-like structures among the positively stained structures (as observed in ref 28) but not in the intestine.

Considering your remark that the structures mentioned as the centrosomes may be rather midbody remnants and considering that centrosomes do not stain for acetylated-tubulin, we have modified the text to consider this (legend of Fig. 1, lines 207-209).     

Remark:

Fig. 3: Again, the staining with a ciliary marker (in this case Arl13b) shows dotted rather than elongated structures. The structures appear to be located apically, near the tight junctions, a location that is typical for midbody remnants. As in Figs. 1 and 2, further characterization of these structures is needed.

Response:

The 10-days requested by the Editor for sending the revised manuscript would not allow us to perform the experiments required to distinguish whether some of the positive structures identified in Fig. 3 are primary cilia or their precursors as midbody remnants. We therefore modified the text to consider this possibility as follows: "It is noteworthy that some of the stained structures for ARL13B could be midbody remnants, a type of structure reported in polarized cells and proposed to be involved in the formation of primary cilia [59]. Interestingly, the model proposed would imply that only cells with a midbody remnant could assemble primary cilia [59], supporting the idea that at least some intestinal BMI1 positive cells could become ciliated." (lines 230-234)

Thank you

Reviewer 3 Report

In the present manuscript, entitled "Primary cilium identifies a quiescent cell population in the human intestinal crypt” Senicourt and colleague claim that they identified a population of cells in the intestine epithelium, in the crypt, that harbor primary cilia when most other cell types are known to be devoid of PC. 

The article’s structure is well built in terms of number of paragraphs and relative content and is globally well-written, and I think this is an important work in the field as we lack a good description of ciliated cell types in the GI tract. This being said, although the science in the paper is pretty simple,  there remains a few questions regarding the data and the manuscript :

Major issues

1)     In many instances throughout the manuscript, the authors claim that a cell is only able to grow a primary cilium when quiescent (for instance, lines 53, 266) which prompt them to test if they are cycling by using BrdU. I think that there is a misunderstanding here because this is incorrect. Most cell types do grow a cilium while cycling. The cilium is resorbed before mitosis to allow the centrioles to organize the mitotic spindle, and the cilium grows back after mitosis. A cell do not need to be quiescent (in G0) to grow cilium. It is true, though, that when a cell population is differentiated, the number of ciliated cells is higher because in an unsynchronized population of cycling cells, the number of cells being in a step of the cell cycle where the cilium is present is fairly low compared to cells in G0 where the presence of cilium is more “stable”. There are many examples where cycling cells will harbor cilia in large portion of their population, ranging from cancer cells (See Eguether and Hahne 2018) to stem cells (See Yanardag and Pugacheva 2021), so please remodel aforementioned statements as well as paragraph 51-60.

2)     In figure 1, with that resolution and only using acetylated tubulin, I don’t believe the authors have ground for claiming that they detect centrosomes. I would remove this from the manuscript, or strengthen this point with at least a co-staining with gamma tubulin or any other marker more specific for centrioles.

3)     In figure 2, although the authors are doing the right thing here by double-checking with Arl13b the reality of their acetylated tubulin staining, this specific picture is really underwhelming. Usually, people tend to publish their best, most convincing picture. If this is what the authors did, this is questioning the fact that they see primary cilia at all! Please provide a more convincing series of images

4)     In Figure 4, the authors mentions in the text that HIEC-6 cells grow undifferentiated in culture but can be induced toward a differentiated phenotype by different means. It is unclear to me if the authors did anything to the cells to induce their differentiation. Please clarify. On the same line of thought, since HIEC-6 cells are grown to confluence in this experiment, does it affect its differentiation state?

5)     In figure 5C, the figure legends states that the experiment has been done 3 times, and yet there is no error bars or associated statistics whatsoever. Please provide associated statistics

6)     In figure 7, the authors do not indicate if the cells are use SC or at any stage of confluence. Please specify.

Also, a 2-fold increase of Gli1 is pretty weak, compared to what is seen in other cell types (MEF, for exemple). It is not uncommon as other cell types are also difficult to activate (like MDCK cells), but authors should comment on this in the manuscript.

Minor issues

1)     Introduction about the link between Hh and cilia (line 39) should include the landmark paper of Huangfu  : Huangfu D, Liu A, Rakeman AS, Murcia NS, Niswander L, Anderson KV (2003Hedgehog signalling in the mouse requires intraflagellar transport proteinsNature 4268387

2)     Line 135 : “with a” is written twice

3)     Line 190 :  “BaseD on”

4)     In all the figures with IF experiments, please add markers on the picture as it was done in Fig1 but nowhere else. This makes reading easier.

Also, please use another terminology for acetylated tubulin as “a-tub” is confusing for the reader who might think this is alpha tubulin.

5)     Line 251 : I believe Acet-Tubuline is only in panel D not AB

6)     Throughout the manuscript, please refrain from using the terminology “primary cilia expression”. It’s becoming a trend in cilia paper to say that, but it makes no sense, as the cilium is the result of the expression of hundreds of genes… Would you write that the cell expresses the mitochondria?

Author Response

Thank you for your positive remarks and suggestions.

Remarks:

1)     In many instances throughout the manuscript, the authors claim that a cell is only able to grow a primary cilium when quiescent (for instance, lines 53, 266) which prompt them to test if they are cycling by using BrdU. I think that there is a misunderstanding here because this is incorrect. Most cell types do grow a cilium while cycling. The cilium is resorbed before mitosis to allow the centrioles to organize the mitotic spindle, and the cilium grows back after mitosis. A cell do not need to be quiescent (in G0) to grow cilium. It is true, though, that when a cell population is differentiated, the number of ciliated cells is higher because in an unsynchronized population of cycling cells, the number of cells being in a step of the cell cycle where the cilium is present is fairly low compared to cells in G0 where the presence of cilium is more “stable”. There are many examples where cycling cells will harbor cilia in large portion of their population, ranging from cancer cells (See Eguether and Hahne 2018) to stem cells (See Yanardag and Pugacheva 2021), so please remodel aforementioned statements as well as paragraph 51-60.

Response:

The sentence in line 266 (now 270-271) has been deleted and the sentences in lines 51-60 have been rewritten as follows: "(...) since the basal body from which the cilium grows needs to be converted into the centrosome required for the mitotic spindle in proliferative cells [25] but in fact, only cells under the mitotic process cannot grow a primary cilium [1,4,26]. Nevertheless, the transit amplifying crypt cells and the primordial stem cells from which they are derived appear not able to assemble primary cilia. Rare reports from detailed electron microscopy (EM) observations in the mouse intestine have clearly established that the terminally differentiated intestinal epithelial cells of the villus, which are quiescent, do not generate primary cilia [26]."

Remark:

2)     In figure 1, with that resolution and only using acetylated tubulin, I don’t believe the authors have ground for claiming that they detect centrosomes. I would remove this from the manuscript, or strengthen this point with at least a co-staining with gamma tubulin or any other marker more specific for centrioles.

      Response:

      As suggested, we removed the term centrosome in the legend of Fig. 1. As suggested by another reviewer, we are considering these structures as midbody remnants.

Remark:

3)     In figure 2, although the authors are doing the right thing here by double-checking with Arl13b the reality of their acetylated tubulin staining, this specific picture is really underwhelming. Usually, people tend to publish their best, most convincing picture. If this is what the authors did, this is questioning the fact that they see primary cilia at all! Please provide a more convincing series of images

      Response:

      The difficulty with the identification of the primary cilium in the intestine is its rarity. As mentioned in the text, on average, only one cilium was detected for each 15 crypts examined. Even in colorectal tumours where primary cilia are observed more frequently (please see ref. 28), the typical elongated structure seen in vitro, on colorectal cancer cells or as here with HIEC-6 cells, was only seen on a few occasions as shown in Fig 1C, using reconstituted deconvoluted images. In other tissues such as the nephron, where a large proportion of the epithelial cells display primary cilia, it is possible to distinguish a few of these elongated-like structures among the positively stained structures (as observed in ref 28) but not in the intestine. We realize that this set of pictures may not be considered as outstanding but we chose to present this figure as a representative illustration of what other investigators interested in this would be likely to observe if performed under similar conditions. 

Remark:

4)     In Figure 4, the authors mention in the text that HIEC-6 cells grow undifferentiated in culture but can be induced toward a differentiated phenotype by different means. It is unclear to me if the authors did anything to the cells to induce their differentiation. Please clarify. On the same line of thought, since HIEC-6 cells are grown to confluence in this experiment, does it affect its differentiation state?

Response:

As suggested, we have added the following sentence to clarify these questions: " Here, using normal non-stimulated cell culture conditions under which the HIEC-6 cells have no ability to differentiate [44,46]" lines 254-255.

Remark: 

5)     In figure 5C, the figure legends states that the experiment has been done 3 times, and yet there is no error bars or associated statistics whatsoever. Please provide associated statistics

Response:

As mentioned in the legend, 3 separate experiments were performed but the times after passage were too different to be compiled and analyzed. We therefore chose to illustrate one of the experiments as representative.

Remark: 

6)     In figure 7, the authors do not indicate if the cells are use SC or at any stage of confluence. Please specify.

      Response:

      As now indicated in the legend, cells were at post-confluence.

      Remark:

Also, a 2-fold increase of Gli1 is pretty weak, compared to what is seen in other cell types (MEF, for exemple). It is not uncommon as other cell types are also difficult to activate (like MDCK cells), but authors should comment on this in the manuscript.

Response:

We did not find specific information about GLI1 inducibility in MDCK cells but variations observed were in the same range as those observed between colorectal cancer cell lines. We thus added the sentence: "Variations in GLI1 and PTCH1 levels were consistent with those observed between the various colorectal cancer cell lines [26].", lines 318-319.

Minor issues

1)     Introduction about the link between Hh and cilia (line 39) should include the landmark paper of Huangfu  : Huangfu D, Liu A, Rakeman AS, Murcia NS, Niswander L, Anderson KV (2003) Hedgehog signalling in the mouse requires intraflagellar transport proteins. Nature 426: 83–87

Response: The reference was added (ref 12), line 39.

2)     Line 135 : “with a” is written twice

      Response: deleted

3)     Line 190 :  “BaseD on”

      Response: corrected

4)     In all the figures with IF experiments, please add markers on the picture as it was done in Fig1 but nowhere else. This makes reading easier.

Also, please use another terminology for acetylated tubulin as “a-tub” is confusing for the reader who might think this is alpha tubulin.

Response: We did. We also changed a-tub for A-a-tub.

5)     Line 251 : I believe Acet-Tubuline is only in panel D not AB

      Response: Exactly. Corrected

6)     Throughout the manuscript, please refrain from using the terminology “primary cilia expression”. It’s becoming a trend in cilia paper to say that, but it makes no sense, as the cilium is the result of the expression of hundreds of genes… Would you write that the cell expresses the mitochondria?

      Response: Replaced in lines 64, 259, 267 and 273.

Thank you

Round 2

Reviewer 1 Report

I still have concerns concerning the photos presented in Fig1 and 2. Cilia are difficult to visualize and additional analyses (immuno using pericentrin or eletronic microscopy) would increase the strength of the paper. I understand that the 10-days requested by the Editor for sending the revised manuscript would not allow the authors to perform the experiments required.

Author Response

Thank you for your positive remarks and suggestions. As agreed with the Editor, we had a significant extension for reviewing the manuscript and, hopefully, fulfilling your concerns.

Remark:

I still have concerns concerning the photos presented in Fig1 and 2. Cilia are difficult to visualize and additional analyses (immuno using pericentrin or eletronic microscopy) would increase the strength of the paper. I understand that the 10-days requested by the Editor for sending the revised manuscript would not allow the authors to perform the experiments required.

Response:

The difficulty with the identification of the primary cilium in the intestine is its rarity. As mentioned in the text, on average, only one cilium was detected for each 25-30 crypts examined on 2-3 micrometer thick sections. In this context, TEM was not possible considering the odds on ultra-thin sections.

However, we have found that one of the problems was that the ARL13B antibody was detecting other structures in addition to the primary cilium on intestinal sections. Thus, as for acetylated tubulin co-staining (Fig. 2), we used co-staining with an antibody against polyglutamylated tubulin and confirmed that only the very few dots identified with ARL13B 17711-1-AP were co-stained (Fig. 3). We observed the same 17711-1-AP antibody was also cross-reacting with centrosome-like components as illustrated in Fig. 4F under our staining conditions on tissues. It is noteworthy that the same antibody was very specific for the cilia on cells (Fig. 11).

We thus repeated the experiments using another anti-ARL13B (90413h from BiCell) and found that this antibody generally stained a single predominant dot in one out of every 25-30 crypts. As suggested, we did co-staining with pericentrin and found that the primary cilium is adjacent to the centrosome, as expected. We also performed co-staining with MKLP1 and found that MKLP1 positive components are frequently expressed in association with the primary cilium. Finally, we also repeated the experiments with BMI1 to confirm that the primary cilium is detected in BMI1 positive cells. 

Reviewer 2 Report

My concerns have been adequately addressed with text modifications.

Author Response

Thank you for your positive remarks. As agreed with the Editor, we had a significant extension for reviewing the manuscript.

Although you indicate that you were satisfied with the corrections in the first revision cycle, we tested the possibility that the primary cilium identified in the intestinal crypt cells could be a midbody remnant, as suggested.  As you can see in Fig. 7 and 8, we tested this possibility and found that the primary cilium identified with a new anti-ARL13B does not co-stain with MKLP1 but both structures are frequently found side by side suggesting a functional link between the primary cilium and the midbody remnant.      

Reviewer 3 Report

To me, the manuscript improved, but not sufficiently to reach publication. None of the experimental data asked (for some of them, by the three reviewers!!! ) were performed. The authors can't settle for cosmetic changes instead of addressing the fundamental issues of the paper. 

To be more factual I would ask, a minima

To see another set of images stained with either acetylated tubulin or Arl13b  (or any other ciliary marker for that matter) showing the BMI + ciliated cells. I understand that the authors want to show something representative and that these cells are rare, but this is the fundamental observation of the paper and if this is not convincing for the cilia community, the paper will be disregarded by most (note to the EDITORS, as this seems to be the only thing that matters these days : this means not cited!!!). 

To either remove the panel showing "centrosome or midbody remnants" or to actually show double staining with a marker for either centrosome or midbodies (or both, one can dream...). The authors cannot simply state "these are X" without actually proving it. For what we know at this point, this could be background from secondary antibodies... Also, contrary to acet-tubulin, I have yet to see, a midbody stained by Arl13b (and I've used 3 different Arl13b antibodies), so I doubt that you can base your assumption on this staining (line 230-231)

Please proof read and rephrase your modified paragraph (lines 53-56) as the ending is actually  contradicting the beginning. It doesn't make much sense to me. 

Second note to the EDITORS : I'm actually a little puzzled by the fact that the authors had only 10 days to respond to 3 reviewers asking collectively for more data, that obviously take much more time to produce. If this is where we are arrived in terms of review process at MDPI, there is no need to get paper reviewed anymore, just change your name to bioRXiv... 

Author Response

Thank you for your remarks and suggestions. As agreed with the Editor, we had a significant extension for reviewing the manuscript and, hopefully, fulfilling your concerns.

Remarks:

To me, the manuscript improved, but not sufficiently to reach publication. None of the experimental data asked (for some of them, by the three reviewers!!! ) were performed. The authors can't settle for cosmetic changes instead of addressing the fundamental issues of the paper.

To be more factual I would ask, a minima :

To see another set of images stained with either acetylated tubulin or Arl13b  (or any other ciliary marker for that matter) showing the BMI + ciliated cells. I understand that the authors want to show something representative and that these cells are rare, but this is the fundamental observation of the paper and if this is not convincing for the cilia community, the paper will be disregarded by most (note to the EDITORS, as this seems to be the only thing that matters these days : this means not cited!!!).

To either remove the panel showing "centrosome or midbody remnants" or to actually show double staining with a marker for either centrosome or midbodies (or both, one can dream...). The authors cannot simply state "these are X" without actually proving it. For what we know at this point, this could be background from secondary antibodies... Also, contrary to acet-tubulin, I have yet to see, a midbody stained by Arl13b (and I've used 3 different Arl13b antibodies), so I doubt that you can base your assumption on this staining (line 230-231)

Please proof read and rephrase your modified paragraph (lines 53-56) as the ending is actually  contradicting the beginning. It doesn't make much sense to me.

Response:

As suggested for the co-staining of ARL13B and BMI1 positive cells, we have added another example of ARL13b identified with the 17711-1-AP antibody with BMI1 positive nuclei (Fig. 4 G-I). In the original panels of Fig. 4, also in Fig. 2 showing acetylated-alpha-tubulin with ARL13b co-staining and a newly added Fig. 3 showing polyglutamylated tubulin with ARL13b, it was found that this antibody cross reacts with non-cilia components under the conditions used. This is a phenomenon that has been reported in the past on tissue sections as compared to cells in culture. Incidentally, in intestinal crypt cells in culture, the same antibody specifically identified the primary cilia as shown in Fig. 11. As hypothesized in the initial manuscript, these structures could be related to midbody remnants (Fig. 4E) and centrosomes (Fig. 4F), based on their locations. As you suggested, we did additional double immunofluorescence staining using anti-pericentrin and anti-MKLP1 rabbit antibodies to identify these structures but had to use another antibody, 90413h, a rat antibody for ARL13b. As shown in the new Fig. 6-8, co-labeling with these new antibodies confirmed our initial observations that only one cilium is normally detected per crypt on average in positive crypts (one out of 15-20 crypts), that the ARL13b positive structure is identified adjacent to a centrosome (pericentrin positive) as expected, as well as to the midbody remnants (MKLP1 positive) supporting a potential close link between the primary cilium and the  midbody remnant in the human intestinal epithelium.       

As also requested, we repeated the experiments with BMI1 with the new anti-ARL13b antibody and new BMI1 antibody to support our initial observation of Fig. 4 and confirmed that the primary cilium detection appears to be associated with BMI1 positive cells (Fig. 9). 

Finally, the phrase in lines 53-60 was rewritten.

Round 3

Reviewer 1 Report

Most of the comments have been addressed. New data have been generated. Figure 9 is not clear to me (BMI staining could be improved).

Author Response

Remark:

Most of the comments have been addressed. New data have been generated. Figure 9 is not clear to me (BMI staining could be improved).

Response:

Considering that BMI1 immunodetection on tissue section from normal human intestinal has never been reported to our knowledge, it was kind of a challenge to do it in co-staining with ARL13b for the primary cilium. Obviously, these components were easier to detect in cells in culture as in HIEC cells.

As mentioned in the newly added text (lines 540-550): " It is noteworthy that the identification of BMI1 as a marker for the +4 reserve stem cells was obtained through various strategies such as the use of reported gene expression system in experimental animal models [35,42,77] while in the human, the BMI1 protein was mainly detected in gastrointestinal cancer cells where it is overexpressed but not or only weakly in their normal counterparts [78-80]. In this study, immunodetection of the BMI1 protein in the human intestine showed discrete cells or cell clusters in the lower crypt region that were stained more intensively than their surroundings showing weak but consistent positive staining, an observation in agreement with the fact that BMI1 while mainly expressed by +4 reserve stem cells also display a broader expression pattern in other intestinal epithelial cells [77,81,82]." 

Hoping these explanations will fulfill your concerns.

Thank you,

Reviewer 3 Report

Thank you for integrating a new set of data as the quality of the paper has improved a lot consequently. The description of primary cilia is now compelling and the systematic colocalization with the midbody remnants of great interest. I believe the paper is now sound and ready for publication.

Author Response

Thank you